# Molecular Mechanisms of Epithelial–Mesenchymal Transition in Retinal Pigment Epithelial Cells: Implications for Age-Related Macular Degeneration (AMD) Progression

**DOI:** 10.3390/biom15060771

**Published:** 2025-05-27

**Authors:** Na Wang, Yaqi Wang, Lei Zhang, Wenjing Yang, Songbo Fu

**Affiliations:** 1The First Clinical Medical College, Lanzhou University, Lanzhou 730000, China; wangna2021@lzu.edu.cn (N.W.); wangyaqi19@lzu.edu.cn (Y.W.); leizhang21@lzu.edu.cn (L.Z.); yangwenjing21@lzu.edu.cn (W.Y.); 2Department of Endocrinology, The First Hospital of Lanzhou University, Lanzhou 730000, China; 3Gansu Province Clinical Research Center for Endocrine Disease, Lanzhou 730000, China

**Keywords:** age-related macular degeneration, epithelial–mesenchymal transition, retinal pigment epithelium, oxidative stress, TGF-β, autophagy

## Abstract

Age-related macular degeneration (AMD), the leading cause of irreversible blindness worldwide, represents a complex neurodegenerative disorder whose pathogenesis remains elusive. At the core of AMD pathophysiology lies the retinal pigment epithelium (RPE), whose epithelial–mesenchymal transition (EMT) has emerged as a critical pathological mechanism driving disease progression. This transformative process, characterized by RPE cell dedifferentiation and subsequent extracellular matrix remodeling, is orchestrated through a sophisticated network of molecular interactions and cellular signaling cascades. Our review provides a comprehensive analysis of the molecular landscape underlying RPE EMT in AMD, with particular emphasis on seven interconnected pathological axes: (i) oxidative stress and mitochondrial dysfunction, (ii) hypoxia-inducible factor signaling, (iii) autophagic flux dysregulation, (iv) chronic inflammatory responses, (v) complement system overactivation, (vi) epigenetic regulation through microRNA networks, and (vii) key developmental signaling pathway reactivation. Furthermore, we evaluate emerging therapeutic strategies targeting EMT modulation, providing a comprehensive perspective on potential interventions to halt AMD progression. By integrating current mechanistic insights with therapeutic prospects, this review aims to bridge the gap between fundamental research and clinical translation in AMD management.

## 1. Introduction

Age-related macular degeneration (AMD), a common chronic degenerative disease, represents the principal cause of visual impairment and vision loss in developed countries. Notably, approximately 7–8% of the global population is afflicted with AMD, and regrettably, up to 90% of these patients currently lack a curative treatment [1]. With the aging population, the number of AMD patients is projected to soar to 288 million by 2040, posing a significant public health challenge [2]. Epithelial–mesenchymal transition (EMT) is a biological process wherein differentiated epithelial cells undergo dedifferentiation to a mesenchymal state. Given its role as a key driver in tumorigenesis, EMT has emerged as a crucial target for cancer therapeutics [3]. However, research on EMT in ocular diseases has been relatively sparse. In recent years, nevertheless, studies have unveiled the essential role of RPE cell EMT in AMD pathogenesis [4]. RPE cells, positioned between photoreceptors and the choroid within the retina, are vital for maintaining photoreceptor health [5]. Under pathological conditions, RPE cells secrete mediators that recruit and activate inflammatory cells, glial cells, and fibroblasts, with the majority of the latter originating from RPE cells themselves [6]. Dysfunction of RPE cells is frequently regarded as the epicenter of AMD etiology. Upon damage or senescence, RPE cells commonly initiate EMT [7], whereby they relinquish intercellular adhesion and apical polarity and acquire mesenchymal traits [8], ultimately culminating in RPE fibrosis, a hallmark of advanced AMD. In light of the compelling evidence highlighting the centrality of EMT in RPE cells, novel therapeutic avenues for AMD have emerged from the EMT perspective. For instance, the transforming growth factor beta (TGF-β) family wields a dominant regulatory role in AMD-related EMT [9]. Moreover, numerous studies have observed the concurrent inhibition of epithelial markers and the activation of mesenchymal markers during EMT [10], spurring interest in the reversibility of EMT and the mesenchymal–epithelial transition (MET) [11]. Thus, activating the MET process and restoring the epithelial functional monolayer could potentially offer a promising novel strategy against AMD.

Advanced AMD, including geographic atrophy (GA, also known as dry or non-exudative AMD) or neovascular AMD (nAMD, also known as wet or exudative AMD), can lead to severe and permanent visual impairment [12] (Figure 1). Dry AMD is characterized by an increase in drusen and extracellular deposits, concurrent with a decline in retinal pigment epithelial (RPE) cells, photoreceptors, and choroidal capillaries [13]. Constituting 70–90% of all AMD cases, dry AMD urgently demands clinical intervention, yet effective remedies remain elusive [14]. It has been suggested that in dry AMD, lysosomal dysfunction may drive RPE cells into the EMT in order to survive the stressful microenvironment. New evidence provided by Ghosh et al. suggests that the βA3/A1-crystallin protein encoded by the Cryba1 gene becomes a potential therapeutic target for dry AMD through the restoration of lysosomal dysfunction and the potential reversal of EMT [11]. Thus, this may provide a novel and effective approach to preventing or delaying the progression of AMD, which deserves to be thoroughly investigated in the future. In recent years, with the development of single-cell technology, researchers have gradually recognised that the lack of patient-derived RPE datasets severely limits the in-depth resolution of RPE pathogenesis, especially at the single-cell resolution level. Xu et al. integrated a large number of RNA-seq datasets from dry AMD in order to extract molecular features of RPE in the pathogenesis of dry AMD. It was found that carboxypeptidase X (CPXM2) showed specific high expression in RPE cells from dry AMD patients, and the molecule was confirmed to be involved in regulating the EMT process in RPE cells by ex vivo experiments. Furthermore, the silencing of CPXM2 suppressed the mesenchymal phenotype of RPE cells in an oxidative stress cell model [15]. Thus, this may represent a potential therapeutic direction for dry AMD. In contrast, wet AMD is typified by choroidal neovascular (CNV) membranes, accompanied by typical lesions such as RPE cell detachments or subretinal fibrous scar tissue [16]. CNV is classified into the following types based on anatomical histopathological location: type 1 CNV is located below the RPE; type 2 CNV occurs between the retina and the RPE; and type 3 neovascularisation begins within the retina and then reaches the subretinal or sub-RPE region. The classic CNV in wet AMD is type 2 CNV [17]. Currently, anti-vascular endothelial growth factor (VEGF) therapy is the frontline treatment for CNV [18]. However, a high incidence of subretinal scar formation (fibrosis) post-treatment can precipitate permanent macular visual system dysfunction [19]. Therefore, there is an urgent need to advance our understanding of subretinal fibrosis and its pathogenic mechanisms in order to develop alternative treatment options. A recent study using anti-VEGF drugs, including bevacizumab (Bev), ranibizumab (Ran), and abciximab (Afl), demonstrated that blockade of the VEGF signalling pathway induced EMT in the RPE, accompanied by the induction of CCN2, a potent pro-fibrotic factor. And further studies showed that co-treatment with CCN5, an anti-fibrotic factor that downregulates CCN2 expression, prevented this adverse effect. Based on the results of this study, the researchers suggest that co-administration of anti-VEGF drugs with CCN5 would be safer. This study lays the groundwork for the development of a highly effective and safer treatment for nAMD [20].

This article comprehensively reviews the principal mechanisms underlying EMT in AMD development, encompassing oxidative stress, hypoxia, inflammation, autophagy, complement activation, and microRNAs. It also scrutinizes the roles of both native and newly generated molecules in the AMD process from an extracellular matrix (ECM) perspective. Additionally, the article delves into the induction mechanisms of the TGF-β family and other signaling pathways pertinent to EMT/AMD. Understanding the contributions of signaling pathway molecules like TGFβ2 in promoting RPE cell EMT will prove invaluable for diagnosing, managing, and devising new biomarkers for drug development targeting retinal fibrosis diseases. Finally, we encapsulate the current state-of-the-art therapeutic strategies targeting EMT in AMD.

## 2. Molecular Mechanisms Underlying EMT

EMT represents the biological process by which epithelial cells transmute into mesenchymal phenotype cells following a specific program. This phenomenon naturally transpires across diverse tissue types and developmental phases [21]. During EMT, epithelial cells forfeit their intracellular junctions and detach from neighboring cells, attaining mesenchymal-like attributes. Consequently, they display robust motility, invasiveness, and a lack of apical polarity [21,22]. Characteristic alterations encompass the loss of epithelial cadherin and an augmented expression of mesenchymal markers, such as N-cadherin, vitamin A, and fibronectin [23]. Based on its biological traits, EMT can be categorized into three distinct types. Notably, type II EMT bears a close association with ocular diseases. It predominantly activates and augments pathological tissue fibrosis, playing a pivotal role in wound healing, tissue repair, and inflammation [24]. Moreover, investigations have revealed that EMT expedites the fibrosis process by generating epithelial-derived fibroblasts accountable for extracellular matrix (ECM) deposition and tissue scar formation [25]. This indicates that ECM is progressively emerging as a focal point in AMD treatment research. Current studies suggest that pathological circumstances like inflammation, wound healing, and cancer can trigger EMT, potentiate the migratory capacity of epithelial cells, and boost the production of ECM components [26]. Some nascent research demonstrates that transcriptional or post-transcriptional regulatory networks, such as miRNA, can also modulate the progression of EMT [27]. Additionally, multiple EMT activators can be actuated via the TGF-β/SMAD, Wnt/β-Chain proteins, and JAK/STAT pathways to govern EMT [27]. Among them, TGF-β stands as a crucial fibrogenic factor, and its induction of EMT is chiefly mediated by the SMAD signaling pathway [28]. The surfacing of more novel mechanisms will assuredly proffer novel concepts and targets for the treatment of assorted eye diseases.

## 3. Molecular Drivers and Regulatory Mechanisms of EMT in AMD

We conducted a comprehensive review of the evidence and molecular driving factors associated with EMT in the context of AMD, with a particular focus on RPE cells and the extracellular matrix. This endeavor aims to foster a more profound comprehension of the mechanisms underlying the occurrence and development of AMD.

### 3.1. Oxidative Stress: A Potent Driver of the EMT/AMD Cascade

The retina stands out as one of the human body’s tissues with the highest oxygen consumption rates, rendering it highly susceptible to the generation of elevated levels of reactive oxygen species (ROS) [29]. Notably, ROS represents the principal contributor to oxidative stress [30]. Research findings have illuminated that ROS can activate TGF-β1 and instigate the EMT process via the PI3K/AKT and MAPK pathways [30]. Keratin 8 (KRT8) is an epithelial marker protein. These studies have demonstrated that oxidative stress induces the upregulation of MAPK1-mediated KRT8 and its phosphorylated forms within RPE cells. Remarkably, the phosphorylation of KRT8 can also precipitate EMT in RPE cells under oxidative stress conditions. Consequently, inhibiting KRT8 phosphorylation holds the potential to enhance the resistance of RPE cells to oxidative stress and forestall their degeneration, thereby presenting a promising therapeutic avenue for AMD [31]. PIWIL4, a human Argonaute-like protein, has been found to trigger epigenetic silencing failure and subsequent AluRNA accumulation in the nucleus under prolonged oxidative stress. Intriguingly, the accumulation of Alu RNA correlates positively with the progression of EMT in RPE cells, leading to the breakdown of tight cell junctions and ensuing cell migration [32]. Interestingly, this experimental study also observed a decrease in Alu transcript levels during the early stages of ROS exposure, suggesting that this period may represent a critical window for AMD treatment interventions. However, further research is warranted to validate this hypothesis. Manganese superoxide dismutase (MnSOD) emerges as a pivotal molecule in the battle against oxidative stress [33]. Experimental evidence has shown that MnSOD can markedly reduce intracellular ROS levels, and its antioxidant prowess aids in suppressing EMT in RPE cells [34]. Additionally, it was discovered that MnSOD can impede the expression of Snail, the principal transcription factor driving EMT in RPE cells [35]. Conversely, Snail can downregulate the expression and activity of MnSOD. Thus, the Snail–MnSOD axis, formed through their reciprocal interactions, may constitute a novel target for AMD prevention [34]. Smad ubiquitination regulatory factor-1 (Smurf1), a ubiquitin-protein ligase, has been the focus of studies. These investigations have revealed that inhibiting Smurf1 not only mitigates ROS-induced damage to RPE cells and the retina but also precludes RPE cells from entering the EMT state. This implies that inhibiting Smurf1 could potentially open new doors for the treatment of dry AMD [36]. PTEN, a tumor suppressor factor, has been shown to play a crucial role. PTEN-deficient RPE cells lose their epithelial characteristics and undergo EMT. The accumulation of oxidative stress can induce PTEN inactivation through phosphorylation while simultaneously activating the PI3K-Akt signaling pathway in RPE cells, ultimately leading to AMD. This firmly establishes the necessity of PTEN’s protein phosphatase activity in inhibiting EMT [37]. Kallistatin (KAL) is a secreted protein with antioxidant properties. Findings indicate that KAL can inhibit oxidative stress-induced EMT by downregulating the transcription factor Snail, thereby potentially contributing to AMD treatment strategies [38]. Furthermore, the Nrf2 pathway represents one of the most significant antioxidant defense mechanisms in the human body [39], and it has been implicated in the regulation of EMT in RPE cells [40]. Researchers have determined that the loss of Nrf2 and Peroxisome proliferator-activated receptor γ coactivator-1α (PGC-1α) genes impacts the EMT of RPE cells through oxidative stress [41]. Future research efforts targeting these pathways may yield more precise prevention and treatment modalities for AMD.

### 3.2. Hypoxia: Inducing the EMT Process in RPE Cells

Hypoxia constitutes a stressor associated with oxygen imbalance during ROS production, which prompts the accumulation of hypoxia-inducible factors (HIFs) and consequent alterations in the expression of numerous genes within cells [42]. In patients with ischemic retinal diseases, both HIF-1α and HIF-2α promote retinal neovascularization. However, Shoda et al. further demonstrated that HIF-1α (but not HIF-2α) may be a critical mediator of subretinal fibrosis. Intriguingly, their earlier work revealed that metabolic dysfunction in RPE cells was specifically associated with HIF-2α rather than HIF-1α. Growing evidence suggests that targeted modulation of HIFs could represent an alternative therapeutic strategy with improved safety and efficacy profiles [43].

Hypoxia-inducible factor-1α (HIF-1α) emerges as the principal regulatory factor for maintaining oxygen homeostasis in wet AMD. Xie et al. discovered that hypoxia-induced HIF-1α expression upregulates p53, and p53-induced miRNA-34a transcription inhibits the expression of Klotho. Additionally, HIF-1α inhibition, p53 mutation, miRNA-34a inhibition, and Klotho overexpression have been shown to reduce hypoxia-induced mesenchymal cell markers, indicating that the HIF-1α/p53/miRNA-34a/Klotho axis promotes hypoxia-induced EMT in ARPE-19 cells. This axis also fuels subretinal fibrosis and exacerbates the formation of CNV. Blocking this axis may offer a novel therapeutic approach for wet AMD [44]. Placental growth factor (PGF), a member of the VEGF family, is responsive to hypoxia, which can induce the expression of the PGF gene in RPE cells [45]. A recent study demonstrated that hypoxia combined with exogenous PGF treatment alters the morphological characteristics of cells, promotes cell migration, and drives EMT-like changes in ARPE-19 cells under hypoxia by activating the NF-κB signaling pathway [46]. The inhibition of PGF and the NF-κB signaling pathway may present a promising target for AMD prevention and treatment. However, it should be noted that exogenous PGF alone does not elicit the aforementioned effects in ARPE-19 cells [46]. The transcription factors of the Snail superfamily play a pivotal role in modulating EMT. A study employing an in vitro hypoxia model to explore the mechanism of EMT in RPE cell lines revealed that hypoxia can enhance the expression of Snail and TGF-β2 and trigger EMT in human RPE cells. Moreover, the inhibition of Snail and TGF-β2 could curtail the development of EMT, and their combined silencing is more effective in suppressing EMT than inhibiting either gene alone [47], potentially representing a novel preventive and therapeutic strategy for AMD. The identification of new transcription factors promises to deepen our understanding of the pathogenesis of EMT/AMD, and the regulation or blockade of transcription factor pathways or their interactions may herald a new direction for future treatment.

### 3.3. Autophagy: Orchestrating the EMT Process in RPE Cells

RPE cells typically engage in basal autophagy to uphold cellular homeostasis within the retina [48]. However, autophagy impairment can precipitate RPE damage and fuel the progression of AMD [49]. Additionally, abnormal RPE cells in AMD exhibit morphological traits characteristic of type II EMT, which can be activated by oxidative-stress-induced autophagy and lysosomal dysfunction [24].

PTEN-induced kinase 1 (PINK1), a key protein in the mitochondrial autophagy pathway [50], has been the focus of research. Findings suggest that PINK1 deficiency in early AMD can disrupt mitochondrial autophagy in RPE cells, thereby triggering NFE2L2-dependent novel retrograde mitochondrial nuclear signaling (RMNS) and ultimately inducing EMT [51]. PGC-1α is a principal transcriptional inducer of mitochondrial biogenesis [52]. Research indicates that the loss of PGC-1α leads to significant alterations in the autophagy capacity of RPE cells, ultimately promoting the loss of epithelial characteristics and triggering EMT transformation. This suggests that PGC-1α preserves the function and phenotype of RPE cells by inhibiting autophagy-mediated EMT [52].

C-X-C motif chemokine receptor 5 (CXCR5), regarded as a G protein-coupled receptor protein essential for RPE homeostasis, has been investigated. Studies have found that the absence of CXCR5 abolishes CXCL13/CXCR5 signaling transduction, thereby perturbing the normal autophagy and EMT pathways and culminating in RPE dysfunction [53,54]. Another study determined that CXCR5 maintains RPE homeostasis by sustaining PI3K/AKT signaling and inhibiting FOXO1 activation, thereby regulating the expression of EMT and autophagy dysregulation genes involved in RPE cells. Thus, the lack of CXCR5 undermines RPE cell function by compromising epithelial integrity and autophagy, leading to EMT [55]. Furthermore, the antioxidant cerium dioxide nanoparticles (CeO2-NP) have been shown to inhibit EMT in RPE cells and modulate autophagy through the downregulation of LC3B-II and p62. Researchers posit that the attenuation of EMT by CeO2-NP in RPE cells may be mediated by autophagy regulation [56], potentially representing a potent therapeutic option for AMD.

The Prominin-1 (Prom1) gene, which is integral to the basic structure of photoreceptor disc biogenesis in the eyes, has been the subject of research. Findings support the notion that Prom1 controls RPE homeostasis through autophagy. Researchers have determined that Prom1-KO downregulates cathepsin D (CTSD) activity and disrupts autophagy flux in an mTORC1 transcription factor EB (TFEB)-dependent mechanism, ultimately inducing EMT. Therefore, the Prom1-mTORC1-TFEB signaling axis may represent the core driving force for RPE cells to maintain homeostasis. The pharmacological modulation of mTORC1 TFEB signaling to correct autophagy and lysosomal function may offer a promising therapeutic strategy for AMD [49].

### 3.4. Inflammation: Inhibiting EMT by Protecting RPE Cells

Researchers have detected the immunohistochemical staining of NLRP3-containing inflammasome in the RPE of patients with advanced AMD, indicating the activation of inflammasome in the pathogenesis of AMD [57]. Additionally, pro-inflammatory cytokines have been shown to significantly reduce the mRNA expression of CDH1, a recognized epithelial marker. The diminished expression of CDH1 may signify EMT-like changes. Consequently, RPE cells may trigger dysfunction when exposed to pro-inflammatory cytokines by downregulating the expression of EMT-related genes [58].

Nurr1, a member of the nuclear receptor subfamily of transcription factors renowned for its anti-inflammatory properties [59], has been studied. Research has revealed that the overexpression and activation of NURR1 can modulate the expression of EMT-related genes and proteins in human RPE cells, attenuating the EMT induced by the pro-inflammatory cytokine TNF-α. Thus, targeting NURR1 may hold therapeutic potential for AMD by regulating EMT and inflammation [60].

Wnt, a critical pathway in EMT regulation, has been implicated in compromising the RPE barrier function and precipitating degenerative diseases of the RPE [61]. A study demonstrated for the first time that Wnt5a, which antagonizes the Wnt/β-catenin pathway, downregulates the expression of pro-inflammatory and angiogenic factors such as NF-κB, TNF-α, and VEGF, and reduces the level of Snail by Wnt5a-induced Snail phosphorylation and degradation and inhibition of the TNF-α/NF-κB pathway, thereby increasing the expression of E-cadherin and inhibiting cell migration. In essence, Wnt5a inhibits EMT in human RPE cells [62].

Adrenal medulla 2 (AM2), a member of the calcitonin superfamily with a significant role in alleviating endoplasmic reticulum and oxidative stress via PI3K-AKT and extracellular-related kinase (ERK)-dependent pathways [63], has been investigated. The experiment found that the exogenous administration of AM2 can inhibit the expression of genes related to inflammation, fibrosis, and oxidative stress through the upregulation of Meox2. AM2 can inhibit the progression of subretinal fibrosis by inhibiting EMT in RPE cells, potentially representing an emerging therapeutic target for nAMD [64].

### 3.5. Complement Activation: Promoting Fibrosis Through EMT

The complement system constitutes an essential component of the innate immune system, and a complementary regulatory system exists within the retina [65]. The complement system can be activated through classical, alternative, and lectin pathways. Upon activation, complement fragments including C3a, C5a, and C4a are generated, which actively partake in diverse immune responses [65]. In AMD, complement activation may drive fibrosis by influencing the EMT of RPE cells.

Blance et al. demonstrated that abnormalities in the ECM, such as collagen fiber cross-linking, can precipitate complement-mediated EMT in resident RPE cells and proposed that C3 genetic ablation offers protection against EMT [66]. Another study found that the complement system partially promotes fibrosis through C5a-C5aR-mediated EMT in RPE cells and indicated that C5a-induced EMT in RPE cells involves the canonical TGF-β pathway (SMAD2/3) and the non-canonical GCPR receptor C5aR pathway (ERK1/2) [67]. Intriguingly, this study also uncovered that TGF-β augments the production of C5/C5a in RPE cells and that treating RPE cells with C5a also stimulates the release of TGF-β1 and TGF-β2, suggesting a positive feedback loop between C5a and TGF-β-induced EMT in RPE cells [67]. Additionally, the release of pro-inflammatory mediators induced by C5a was observed, highlighting the interplay between inflammation and complement activation in AMD treatment.

### 3.6. MicroRNA: A Regulator of EMT

MicroRNAs (miRNAs) are small endogenous RNA molecules 19–25 nucleotides in length. They regulate post-transcriptional gene expression and modulate developmental and cellular processes in eukaryotes [68]. In retinal pigment epithelial (RPE) cells, certain miRNAs regulate epithelial–mesenchymal transition (EMT). For example, miRNA-204 and miRNA-211 are highly expressed in differentiated RPE cells and are essential for maintaining RPE characteristics. The inhibition of miRNA-204/211 upregulates EMT-related transcription factors, highlighting their role in suppressing EMT [26,69]. MiRNA-34a, a p53-regulated miRNA, is elevated in wet AMD patients. HIF-1α promotes EMT in ARPE-19 cells by upregulating p53 and miRNA-34a while downregulating Klotho. p53 also enhances fibrosis via miRNA-34a, suggesting its therapeutic potential for wet AMD [44]. MiR-302d and miR-93 target TGFβR2 and VEGFA. They inhibit TGF-β-mediated EMT by disrupting TGF-β signaling in ARPE-19 cells. These miRNAs also reverse TGF-β-induced mesenchymal changes, promoting mesenchymal-to-epithelial transition (MET). Thus, miR-302d or miR-93 overexpression may counteract EMT [70]. MiR-27b-3p directly inhibits TGF-β2-induced EMT in ARPE-19 cells. Exosomes from human umbilical cord mesenchymal stem cells (hucMSC-Exo) containing miR-27b suppress TGF-β-induced EMT by targeting HOXC6, offering a potential wet AMD therapy [71]. Further exploration of miRNAs and their interactions may advance targeted EMT treatments for AMD (Table 1).

### 3.7. Extracellular Matrix (ECM): Balancing Dynamics in EMT of AMD

The Bruch’s membrane, a pentameric ECM structure situated between the retinal pigment epithelium (RPE) and choriocapillaris, provides structural support to the RPE and facilitates nutrient exchange [72]. Age-related changes, such as collagen cross-linking, reduce the elasticity and permeability of Bruch’s membrane, contributing to the susceptibility to age-related macular degeneration (AMD) [66]. These alterations in Bruch’s membrane may influence RPE cell EMT, thereby driving AMD onset and progression [73]. Additionally, the dysregulation of ECM components, including fibronectin, collagen, and matrix metalloproteinases (MMPs), plays a pivotal role in EMT initiation and progression [24].

The disruption of epithelial integrity, marked by reduced E-cadherin expression, serves as a key trigger for EMT in RPE cells [8]. This transition is characterized by the upregulation of mesenchymal markers such as N-cadherin, vimentin, and α-SMA [74]. Transforming growth factor-beta (TGF-β) and connective tissue growth factor (CTGF) stimulate RPE cells to secrete fibronectin (FN), a major ECM component in subretinal fibrosis [75]. At the molecular level, the expression of CTGF is modulated by the Yes-associated protein (YAP) and TAZ protein (YAP/TAZ) complex. Researchers have hypothesized that YAP may mediate mechanotransduction through currently unidentified pathways. Zhang et al. demonstrated that YAP knockdown significantly suppresses TGF-β2-mediated fibrotic and EMT processes in ARPE-19 cells. However, this study primarily focused on a proliferative vitreoretinopathy (PVR) experimental model, and future studies should further characterize YAP’s functional role in AMD experimental models [76]. FN provides a structural scaffold for macrophage and RPE cell migration during EMT [77]. P-cadherin, a critical regulator of cell–cell adhesion, is essential for RPE maintenance. Mutations in P-cadherin lead to RPE atrophy, and its conversion to N-cadherin during EMT underscores its role in RPE cell function and migration [78]. Investigating P-cadherin-mediated cell adhesion mechanisms may offer insights into preventing EMT in AMD [78]. Furthermore, the transcription factor FOXF2 is upregulated during EMT and promotes the expression of E-cadherin inhibitors ZEB1 and ZEB2, further driving EMT progression [79]. Additional evidence demonstrates that TAZ and its coactivator TEAD1 regulate the EMT transcription factor ZEB1 to control RPE cell proliferation and differentiation. Whether the targeted intervention of these pathways can reverse EMT in AMD models warrants further investigation [80].

MMPs can degrade almost all protein components in ECM and influence ECM remodeling and ultimately affect EMT by participating in tissue remodeling, cell migration and invasion, proliferation, and angiogenesis [81,82]. The activation of MMP-1 by lysosomal enzymes in aging and dysfunctional RPE cells leads to the development of late wet AMD in susceptible individuals [24]. MMP-2 is the most abundant enzyme synthesized by RPE cells, and its disrupted activity is a key pathogenic factor in the early development of AMD [83]. An imbalance between MMP-2/9 and their inhibitors, TIMP-1/2, is critical in both early dry AMD and late wet AMD [82]. This imbalance disrupts ECM integrity by degrading key structural proteins, including fibronectin, type IV collagen, type V collagen, and laminin [84]. MMP-3 also directly induces EMT by activating the Rac1 GTPase-ROS signaling pathway, upregulating SNAI1, and promoting the expression of MMP-2 and MMP-9 [85] (Figure 2).

## 4. Roles of Key Cytokine-Mediated Signaling Pathways in EMT/AMD

TGF-β, a member of the growth factor superfamily, is a tightly regulated signaling molecule with complex transcriptional and translational processes. The TGF-β/SMAD pathway [86] is the primary signaling cascade driving EMT in RPE cells, alongside the Wnt/β-catenin [61] and Jagged/Notch [87] pathways. Extensive crosstalk between these pathways creates a complex regulatory network that significantly impacts AMD progression (Figure 3).

### 4.1. TGF-β/SMAD: The Central Signaling Pathway in EMT/AMD

The TGF-β/SMAD pathway regulates EMT through both canonical and non-canonical mechanisms. TGF-β is secreted in a latent form and activated through various mechanisms [88]. In the canonical pathway, activated TGF-β binds to type I and type II serine/threonine kinase receptors, initiating downstream signaling via the phosphorylation of the type I receptor by the type II receptor kinase [89]. Non-canonical pathways, including MAPKs (p38, JNK, ERK) and PI3K/AKT [90], further modulate TGF-β responses.

SMAD proteins are critical downstream mediators of TGF-β signaling [91]. In RPE cells, exogenous TGF-β1 upregulates endogenous TGF-β1 and activates SMAD3-ERK1/2-mTORC1 signaling, promoting ROS production and EMT. The inhibition of this pathway attenuates EMT and restores RPE structural integrity in vitro and in vivo [92]. SMAD7, a negative regulator, inhibits EMT-induced fibrosis by blocking SMAD2/3 phosphorylation [93]. Additionally, Hic-5, a paxillin family member, negatively regulates SMAD3, suggesting a balancing mechanism in EMT [94]. SMAD3 antagonists have shown promise in blocking end-stage AMD [91].

### 4.2. Wnt/β-Catenin: A Collaborative Pathway in EMT/AMD

While Wnt signaling alone does not induce EMT, it synergizes with TGF-β to regulate EMT in AMD. The nuclear import of β-catenin and SMAD3, mediated by LEF proteins, is critical for EMT [95,96]. The β-catenin-dependent activation of LEF-1 induces EMT via the PI3K-Akt pathway [97,98]. Wnt/β-catenin signaling also enhances RPE proliferation and regeneration [99]. The inhibition of Wnt signaling downregulates inflammatory and angiogenic factors, providing insights into AMD pathogenesis [100]. Crosstalk between TGF-β and Wnt pathways, such as LEF-1 activation by β-catenin or SMAD proteins [77], underscores their interconnected roles. Wnt/β-catenin signaling stabilizes SNAI1 phosphorylation through interactions with TGF-β [101], while aberrant Wnt activation [102,103] directly upregulates SNAI1 and SNAI2 [104,105,106]. Targeting Wnt/β-catenin signaling to modulate RPE degeneration represents a promising strategy for wet AMD treatment [99].

### 4.3. Jagged/Notch: A Fibrosis-Related Pathway in EMT/AMD

The Jagged/Notch pathway is a key intracellular mechanism linked to fibrosis and EMT. Notch receptors, upon binding to ligands, undergo γ-secretase-mediated cleavage, releasing the Notch intracellular domain (NICD) into the nucleus to regulate fibrotic gene expression [107]. In TGF-β2-induced EMT, Jagged-1 and Notch-3 are upregulated, along with downstream targets Hes-1 and Hey-1 [87]. The inhibition of Jagged-1 or Notch receptor cleavage blocks TGF-β2-induced EMT by suppressing Snail, Slug, and Zeb1 expression. Notch-1 upregulation in TGF-β2-induced RPE involves both canonical SMAD and non-canonical PI3K/AKT and MAPK pathways [108]. The interplay between ERK1/2, SMAD, and Jagged/Notch signaling highlights the complexity of TGF-β2-induced EMT in RPE. Future research should focus on deciphering the crosstalk between these pathways and developing strategies to modulate their interactions for therapeutic benefit.

## 5. Therapeutic Strategies Targeting EMT in AMD

### 5.1. Inhibiting EMT in AMD Through TGF-β Pathway Modulation

Inhibiting or reversing EMT to prevent subretinal fibrosis represents a promising clinical approach for AMD treatment. TGF-β2 has been a focal point in many studies. For instance, the RAR agonist Am580 attenuates TGF-β2-induced mesenchymal marker expression and inhibits EMT in RPE cells [109]. Similarly, the RAR-γ agonist R667 suppresses TGF-β2-induced MMP release [28]. Further research on these RAR agonists could advance AMD treatment. Pirfenidone inhibits TGF-β2-induced EMT and VEGF secretion in RPE cells by blocking the NF-κB/Snail pathway [110]. Additionally, the IKKβ inhibitor, targeting NF-κB signaling, reverses TGF-β/TNF-α-induced RPE-EMT, restoring RPE identity and offering potential for treating retinal diseases involving EMT [111]. Other inhibitors, such as CCG-1423 (MRTF-A) [112], epoxomicin (proteasome) [113], erlotinib [114], baicalein [114], and sorafenib [22], also suppress EMT by modulating TGF-β signaling, highlighting their therapeutic potential for AMD.

### 5.2. Targeting Oxidative Stress to Inhibit EMT in AMD

ROS production is a key driver of EMT, making antioxidant therapy a viable strategy. Luteolin, a potential dry AMD treatment, protects RPE cells from oxidative damage by promoting Nrf2 nuclear translocation, enhancing antioxidant enzyme expression, and inhibiting AKT/GSK-3β signaling [39]. It also suppresses EMT by inactivating Smad2/3 and YAP pathways, offering anti-fibrotic benefits [115]. Resveratrol reduces oxidative stress and cell proliferation by inhibiting ERK activation [116]. Antioxidants like vitamin E may delay AMD progression, though further clinical trials are needed [117]. Notably, sodium iodate exacerbates ROS production and AMD, but ERK inhibitors like FR180204 can mitigate NaIO3-induced EMT, suggesting a novel therapeutic avenue [118,119].

### 5.3. Reversing EMT in AMD to Restore Epithelial Function

EMT and MET are dynamic processes, and reversing EMT to restore epithelial function is a promising therapeutic strategy. Natural compounds such as curcumin, isoflavone, lycopene, 3,3′-diindolylmethane, indole-3-methanol, and epigallocatechin-3-gallate have shown potential in reversing EMT [120,121]. These agents may offer future treatment options for AMD. In vitro studies have revealed that the inhibition of αB-crystallin can even induce MET, suggesting its potential as a therapeutic target for reversing EMT [122].

### 5.4. Addressing Risk Factors and Cellular Senescence

Cigarette smoke extract (CSE) induces oxidative stress and mitochondrial dysfunction in RPE cells, promoting EMT [123]. Thus, reducing risk factors like smoking and targeting aging RPE cells may help delay AMD progression. In addition, the removal of senescent cells by certain methods may be a promising approach to preventing or delaying the progression of RPE-EMT-associated retinal diseases such as AMD. The anti-aging protein α-klotho inhibits TGF-β2-induced RPE degeneration by suppressing EMT and oxidative stress, offering the potential for dry AMD treatment [124]. Jiang et al. demonstrated that the overexpression of klotho prevented TGF-β1-induced EMT, and in vivo experiments also observed a reduction in subretinal fibrotic areas after klotho treatment, which further suggests that klotho also has an antifibrotic effect on subretinal fibrosis [125]. Recent studies have identified metformin as a promising therapeutic candidate for AMD. Through bioinformatics analysis, Urooba Nadeem et al. demonstrated that metformin shows the strongest genetic association with wet AMD among existing drugs and ranks as a top candidate across all dry AMD subtypes [126]. Mechanistically, Hua et al. revealed that metformin suppresses TGF-β1-induced EMT in ARPE-19 cells and ameliorates laser-induced subretinal fibrosis in murine models by inhibiting miR-126-5p to activate Klotho signaling [127]. These findings collectively highlight metformin’s therapeutic potential for AMD while underscoring the need for further investigation into its precise molecular mechanisms. A study by Gao et al. found that Dasatinib plus quercetin selectively eliminated senescent cells and inhibited low-density TGF β-induced EMT in RPE cells [7]. Intriguingly, a previous study demonstrated that the upregulation of caveolin-1 effectively inhibited EMT in RPE cells and reduced subretinal fibrosis in mice. However, while suppressing fibrosis, caveolin-1 simultaneously promoted RPE cell senescence—a process that may potentially accelerate GA progression in AMD [128]. This finding underscores the need for further investigation into the intricate interplay between RPE senescence and EMT.

### 5.5. Cell Replacement Strategies to Target RPE-EMT in AMD

Cell therapy can halt or reverse AMD by replacing degenerated RPE, thereby restoring retinal function and vision. Clinical trials of RPE transplantation have shown preliminary signs of success: The study by Cruz et al. supports the feasibility and safety of human embryonic stem-cell-derived RPE (hESC-RPE) patch transplantation as a regenerative strategy for AMD [129]. Kashani et al. have developed a clinical-grade retinal implant made of hESC-RPE. In a first-in-human phase 1 clinical trial in five patients with advanced non-neovascular AMD, the implant was shown to be safe and well tolerated [130]. Liu et al. demonstrated that human RPE stem-cell-derived RPE (hRPESC-RPE) transplanted into the subretinal space of non-human primates successfully integrated with the host retina, restored RPE-specific markers, and supported photoreceptor function. Importantly, the transplanted grafts did not undergo EMT [131]. Tian et al. successfully reprogrammed human induced pluripotent stem-cell-derived RPE (iPSC-RPE) into induced RPE (iRPE) cells using a combination of four transcription factors (TFs): CRX, MITF-A, NR2E1, and C-MYC. These critical TFs suppressed EMT in iRPE cells by specifically targeting BMP7 and FOXF2. Notably, the iRPE cells demonstrated not only remarkable resistance to EMT but also superior therapeutic efficacy compared to their parental iPSC-RPE cells [132]. Similarly, Zhu et al. successfully transdifferentiated human umbilical cord mesenchymal stem cells (hUCMSCs) into iRPE cells using five TFs: CRX, NR2E1, C-MYC, LHX2, and SIX6. The resulting iRPE cells exhibited characteristics comparable to iPSC-RPE and demonstrated superior therapeutic efficacy to native hUCMSCs in an AMD rat model. Importantly, these iRPE cells also acquired resistance to EMT [133]. These cells may represent promising clinical candidates for reversing AMD pathophysiology and potentially restoring visual function in the future. Recent studies have demonstrated the critical roles of vitamins and amino acids in regulating RPE function. Shen et al. elucidated that combined treatment with vitamin C and valproic acid activates MET in fetal RPE stem-like cells (fRPESCs) through the SOX2/TGF-β1/SNAIL1 and SOX2/SNAIL1/E-cadherin pathways [134]. This study provides novel insights into graft modification strategies to enhance the efficacy of cell replacement therapies.

### 5.6. Omics Landscapes of RPE-EMT

Ma et al. established the transcriptomic landscape of RPE cells in an nAMD model, identifying CPT1A as a pivotal regulator of metabolic reprogramming and subsequent RPE-EMT. Notably, CPT1A overexpression rescued subretinal fibrosis both in vivo and in vitro, demonstrating its therapeutic potential [135]. Sripathi et al. performed time-course transcriptomic analyses on hRPE monolayers undergoing EMT. The results identified several key regulatory hubs governing RPE-EMT, including axon guidance signaling and other critical kinases, transcription factors, and miRNAs [136]. Another study by this team conducted a comprehensive proteomic analysis to characterize temporal protein expression changes associated with EMT in stem-cell-derived RPE cells. A total of 532 proteins were identified as differentially regulated during RPE-EMT. Integrated analysis of proteomic and prior transcriptomic (RNA-seq) data further revealed the coordinated modulation of multiple AMD-associated risk factors, reinforcing the mechanistic role of EMT in AMD pathogenesis [137]. Metabolomics, defined as the comprehensive analysis of small molecules (metabolites) in biological systems, represents a rapidly expanding field. In a recent study, Huang et al. employed untargeted metabolomics to demonstrate that RPE cells exhibit remarkable metabolic flexibility during EMT induction, capable of rapidly adapting and rewiring metabolic pathways. Their findings revealed that TNFα disrupts metabolic pathways involved in inflammation and oxidative stress, while TGFβ2 impairs the TCA cycle and fatty acid oxidation. These results provide a systems-level understanding of the bioenergetic rewiring processes underlying TGFβ2- and TNFα-dependent EMT induction [4]. Collectively, these studies delineate the multi-omics landscape of RPE-EMT, providing a robust foundation for developing targeted pharmacological strategies to modulate this pathogenic process.

### 5.7. Epigenetic Regulation of RPE-EMT: Implications for AMD Therapy

Growing interest has emerged in epigenetic reprogramming’s role in EMT regulation, owing to its ability to mediate more stable, long-lasting, and reversible cellular modifications [138]. N^6^-methyladenosine (m^6^A), the most abundant epigenetic modification in eukaryotic messenger RNA (mRNA), has recently been implicated in subretinal fibrosis and RPE-EMT through the METTL3-m^6^A-HMGA2 axis [139]. This discovery opens new avenues for therapeutic research. Furthermore, DNA methylation serves as a critical epigenetic mechanism. Studies have revealed that the deficiency of DNA methyltransferase 1 (DNMT1)—predominantly expressed in human RPE cells—significantly impairs photoreceptor differentiation, leading to reduced and mislocalized rhodopsin-expressing cells [140]. Collectively, these findings underscore the pivotal role of DNA methylation in regulating RPE cell fate and survival, positioning it as a promising therapeutic target for future interventions.

## 6. Conclusions and Future Perspectives

EMT is a reversible, multi-step process influenced by oxidative stress, hypoxia, autophagy, inflammation, and complement activation [141]. These mechanisms interact intricately; for example, hypoxia induces autophagy, while oxidative stress stimulates autophagy to mitigate ROS damage [42]. Hypoxia-induced HIF-1α further regulates autophagy and mitophagy, highlighting the interconnectedness of these pathways [142]. In the future, investigating the crosstalk between signaling pathways that regulate RPE-EMT will contribute to the development of novel therapies for AMD.

The molecular regulation of EMT involves transcriptional control, RNA splicing, and miRNA networks [143]. miRNAs play a critical role in maintaining RPE characteristics and regulating EMT-related genes and pathways. ECM remodeling and fibrosis are also central to AMD pathogenesis. Future research should focus on the interplay between EMT transcription factors and ECM proteins, as well as the crosstalk among signaling pathways like TGF-β/SMAD, Wnt/β-catenin, and Jagged/Notch. Emerging studies suggest that transcriptional or post-transcriptional regulatory networks such as miRNAs can also regulate the progression of EMT [27].

This review highlights the role of miRNAs in regulating the EMT by modulating gene expression and pathways essential for maintaining RPE characteristics. Additionally, ECM fibrosis and its associated factors are critical in AMD. The activation of EMT transcription factors by various inducers alters ECM mesenchymal markers, driving EMT. Investigating how EMT transcription factors activate ECM mesenchymal proteins and their interactions represents a key future direction. TGF-β, a central regulator of EMT, activates multiple signaling pathways, including TGF-β/SMAD, Wnt/β-catenin, and Jagged/Notch. However, the crosstalk between these pathways remains underexplored, necessitating further study.

This review has limitations. First, Müller glial cells are the major glial cell type found in the retina and provide support for retinal neurons and the vascular system. Müller glial cells are capable of producing large quantities of ECM proteins and are actively involved in retinal repair and remodeling under certain pathological conditions [144]. However, the limited research on glial cells in EMT restricts comprehensive analysis. Future studies should explore EMT mechanisms in these cell types to enhance AMD therapeutic strategies. Second, the complex interactions between molecules, signaling pathways, and mechanisms require further investigation into intracellular cascades and pathway crosstalk. Clarifying these interactions will provide critical insights into AMD pathogenesis and inform potential treatments.

## Figures and Tables

**Figure 1 biomolecules-15-00771-f001:**
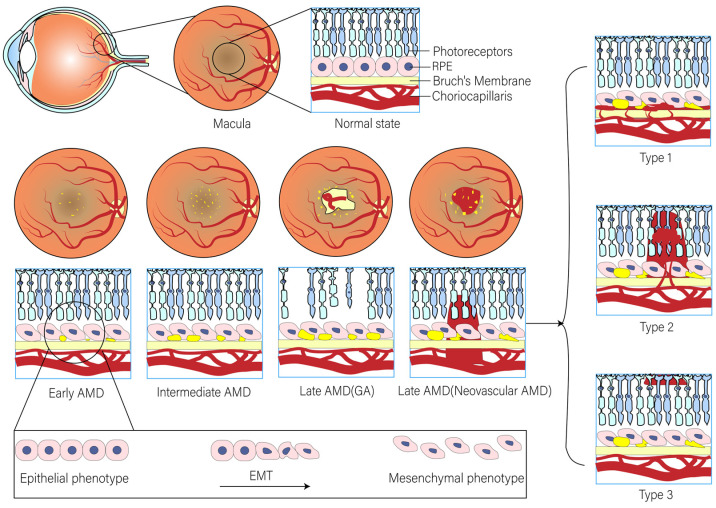
The EMT process of RPE in AMD progression.

**Figure 2 biomolecules-15-00771-f002:**
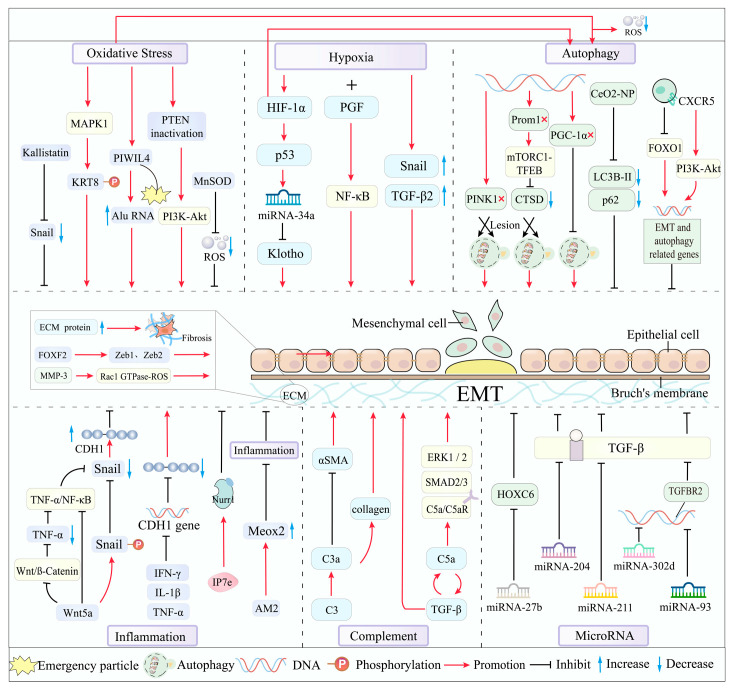
Main mechanisms and molecular drivers of EMT in AMD.

**Figure 3 biomolecules-15-00771-f003:**
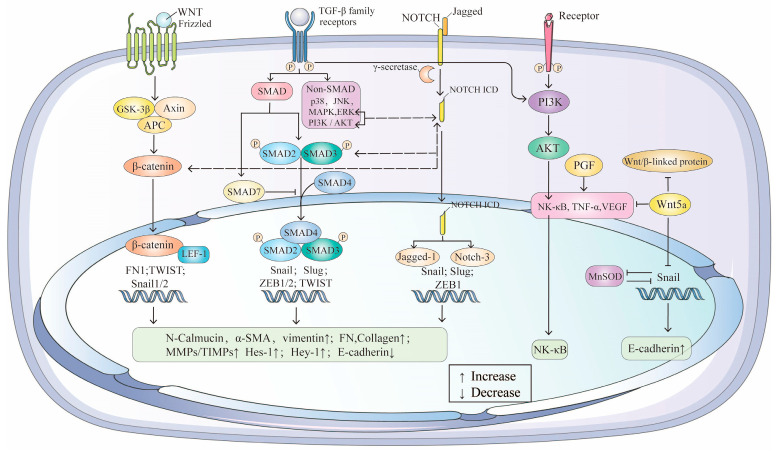
The crosstalk between TGF-β/SMAD, Wnt/β-catenin, Jagged/Notch and NF-κB signaling pathway and in EMT of AMD.

**Table 1 biomolecules-15-00771-t001:** Molecular drivers of EMT in AMD.

Mechanism	Research Object	Study Model	Target Subtype	Possible Pathways	Major Roles	References
Oxidative Stress	KRT8	ARPE-19 cells; human primary RPE cells	N/A	MAPK1/ERK2, MAPK3/ERK1	Induce EMT	[31]
	PIWIL4	ARPE-19 cells	N/A	AKT—Phosphorylation, PIWIL4 chelates into cytoplasmic stress granules	Induce EMT	[32]
	MnSOD	ARPE-19 cells	N/A	Snail-MnSOD axis	Inhibit EMT	[34]
	Inhibit Smurf1	ARPE-19 cells; NaIO_3_-induced *C57BL/6J* mice	Dry AMD	TGF-β pathway and NF-κβ pathway.	Inhibit EMT	[36]
	PTEN deficiency	PTEN^flox^ mice	N/A	PI3K-Akt	Induce EMT	[37]
	Kallistatin	ARPE-19 cells; NaIO_3_-induced *C57BL/6J* mice	Dry AMD	Downregulate the transcription factor Snail	Inhibit EMT	[38]
	Nrf2/PGC-1α deficiency	Double knock-out *C57BL/6J* mice	Dry AMD	Weakened antioxidant defenses caused by lack of two genes, Mitochondria/autophagy deficiency	Induce EMT	[41]
Hypoxia	HIF-1α	ARPE-19 cells; *C57BL/6J* mice CNV model	N/A	HIF-1α/p53/miRNA-34a/Klotho axis	Induce EMT	[44]
	PGF	ARPE-19 cells	Wet AMD	NF-κB	Induce EMT	[46]
	Combined silencing of TGF-β2 and Snail genes	ARPE-19 cells	Wet AMD	knockdown of both inhibited EMT to a greater extent than knockdown of either gene individually	Inhibit EMT	[47]
Autophagy	Loss of Prom1	Isolated RPE cells from *C57/BL6J* mice	Dry AMD	Impaired autophagy; Prom1-mTORC1-TFEB axis	Promote EMT	[49]
	PINK1 deficiency	Human autopsy eyes; *C57BL/6J* mice	Early AMD	RMNS, Nrf2, TXNRD1, PI3K/AKT	Induce EMT	[51]
	PGC-1α deficiency	ARPE-19 cells; *C57BL/6J* mice	N/A	Impaired autophagy	Induce EMT	[52]
	CXCR5 deficiency	*C57BL/6J* mice	N/A	CXCL13/CXCR5, PI3K/AKT/FOXO1 signal axis, Impaired autophagy	Induce EMT	[53,54,55]
	CeO2-NP	ARPE-19 cells; Light-damaged *SD* albino rats	Dry AMD	Interference Autophagy Pathway	Inhibit EMT	[56]
Inflammation	Proinflammatory cytokines	ARPE-19 cells	N/A	Gene Expression Regulation	Induce EMT	[58]
	Nurr1	Primary human RPE cells; ARPE-19 cells; *C57BL/6J* mice	N/A	Regulate the expression of EMT-related genes and proteins	Inhibit EMT	[60]
	Wnt5a	hTERT-PRE-1cells; ARPE-19 cells	N/A	Antagonistic Wnt/β-catenin Pathway, TNF-α/NF-κB	Inhibit EMT	[62]
	AM2	ARPE-19 cells; Laser-induced *C57BL/6J* mice	Wet AMD	Upregulate Meox2, Suppress Inflammation	Inhibit EMT	[64]
Complement Activation	The genetic ablation of C3	iPSC-RPE	N/A		Inhibit EMT	[66]
	C5a	Human eye samples; Laser-induced *C57BL/6J* mice	Wet AMD	Smad2/3, ERK1/2, C5aR	Induce EMT	[67]
MicroRNA	miRNA-204	hfRPE	N/A	Inhibition Of TGF-β Pathway	Inhibit EMT	[26,69]
	miRNA-211	hfRPE	N/A	Inhibition Of TGF-β Pathway	Inhibit EMT	[26,69]
	miRNA-34a	ARPE-19 cells; *C57BL/6J* mice CNV model	N/A	Inhibition Of TGF-β Pathway	Inhibit EMT	[44]
	miR-302d	ARPE-19 cells	Wet AMD	Inhibition Of TGF-β Pathway	Inhibit EMT	[70]
	miR-93	ARPE-19 cells	Wet AMD	Inhibition Of TGF-β Pathway	Inhibit EMT	[70]
	miR-27b	Human skin fibroblasts; ARPE-19 cells	Wet AMD	miR-27b/HOXC6 axis	Inhibit EMT	[71]

N/A indicates that the study did not focus on specific AMD subtypes or the information was not explicitly stated in the original publication.

## Data Availability

Not applicable.

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
