# Peer review of "Molecular Mechanisms of Epithelial–Mesenchymal Transition in Retinal Pigment Epithelial Cells: Implications for Age-Related Macular Degeneration (AMD) Progression"

_biomolecules, 2025, doi:10.3390/biom15060771_

Round 1

Reviewer 1 Report

Comments and Suggestions for Authors

This review article by Na Wang et al. nicely summarizes RPE EMT mechanisms in atrophic AMD pathogenesis and will be helpful to readers in the field. Understanding RPE EMT is essential because it lies at the intersection of retinal disease pathogenesis, RPE dysfunction, and AMD pathology. Although the review is well written, I have the following comments outlined below.

  1. The authors should consider providing additional details on the role of glial cells in EMT and their contribution to AMD pathogenesis. Glial cells (Muller cells) can undergo EMT and contribute to the fibrotic process.
  2. It would be helpful to have a better description of dry and wet AMD and the role of EMT in regulating dry/wet AMD pathogenesis. 
  3. The authors discuss the molecular crosstalk, but it would be beneficial if they could provide a table or summary of the critical molecular drivers of RPE EMT and highlight the crosstalk between the different signaling pathways regulating RPE EMT pathways. Consider adding information related to Yap/TAZ (targeting mechanotransduction) and RUNX family proteins.
  4. Comprehensive pathway cartoons/diagrams/schematics could help summarize the well-characterized models in RPE EMT.
  5. Additional discussions about emerging therapies to prevent/slow RPE EMT, such as small molecule inhibitors, gene therapy approaches, and epigenetic modulation by HDAC and BET inhibitors, would be helpful.
  6. Adding published transcriptomic/proteomic findings during EMT, primarily using iPSC-derived RPE disease models, would be helpful to readers interested in RPE EMT.

Author Response

Dear editors and reviewers:

Thank you for your letter and for the reviewers’ comments concerning our manuscript entitled “Epithelial-Mesenchymal Transition as a Key Player in AMD: Molecular Pathways and Emerging Therapies” ((Manuscript ID: biomolecules-3572242)). Those comments are all valuable and very helpful for revising and improving our paper, as well as the important guiding significance to our researches. We have studied comments carefully and have made correction which we hope meet with approval. Revised portion are marked in red in the paper. The main corrections in the paper and the responds to the reviewer’s comments are as flowing:

Responds to the reviewer’s comments:

Reviewer #1:

Comments 1: The authors should consider providing additional details on the role of glial cells in EMT and their contribution to AMD pathogenesis. Glial cells (Muller cells) can undergo EMT and contribute to the fibrotic process.

Response 1: We appreciate the reviewer’s insightful suggestion regarding the role of glial cells in EMT and AMD pathogenesis. As the reviewer rightly noted, Müller glial cells can undergo EMT and contribute to fibrosis. However, our review primarily focuses on the molecular mechanisms of EMT in retinal pigment epithelial (RPE) cells and their implications for AMD progression, as reflected in our revised title (Molecular Mechanisms of Epithelial-Mesenchymal Transition in Retinal Pigment Epithelial Cells: Implications for AMD Progression). In the Discussion section, we have acknowledged the potential involvement of Müller glial cells (e.g., their ECM production and role in retinal repair under pathological conditions [144]). ( Lines 597-601) Nevertheless, a comprehensive analysis of glial cell-derived EMT remains challenging due to the limited research on glial cells and fibroblasts in this context. Future studies exploring crosstalk between RPE and glial cells during EMT would be valuable to address this gap.

Comments 2: It would be helpful to have a better description of dry and wet AMD and the role of EMT in regulating dry/wet AMD pathogenesis.

Response 2: We appreciate the valuable suggestion to clarify the distinctions between dry and wet AMD pathogenesis and their relationship with EMT. We have significantly expanded the discussion of AMD subtypes and EMT’s regulatory role in Section 1 of the revised manuscript. The text now explicitly distinguishes dry AMD from wet AMD, while emphasizing EMT’s dual contributions to both forms of pathogenesis. For dry AMD, we highlight emerging mechanisms such as CPXM2-mediated EMT regulation in oxidative stress models [15] and Cryba1’s therapeutic potential via lysosomal repair [11]. For wet AMD, we detail subtype-specific CNV classifications and the unintended pro-fibrotic effects of anti-VEGF therapies, proposing combinatorial strategies (e.g., CCN5 co-treatment) to mitigate EMT [20]. (Lines 62-104) To enhance clarity, Table 1 now includes a ‘Target Subtype’ column categorizing cited studies by AMD type (dry/wet/both), ensuring readers can readily contextualize mechanistic findings. These revisions collectively address the intricate interplay between EMT and AMD heterogeneity, while guiding future subtype-targeted therapeutic development.

Comments 3: The authors discuss the molecular crosstalk, but it would be beneficial if they could provide a table or summary of the critical molecular drivers of RPE EMT and highlight the crosstalk between the different signaling pathways regulating RPE EMT pathways. Consider adding information related to Yap/TAZ (targeting mechanotransduction) and RUNX family proteins.

Response 3: We appreciate the reviewer’s insightful suggestions regarding the molecular crosstalk in RPE-EMT. In response, we have (1) added a simplified representation of key crosstalk mechanisms in Figure 2. However, due to the limited systematic research in this area, a comprehensive summary table of critical molecular drivers is currently challenging to compile. (2) As recommended, we have expanded the Discussion section to emphasize the interplay between different signaling pathways regulating RPE-EMT. (Lines 578-579) (3) Additionally, after a thorough literature review, we have incorporated information on Yap/TAZ (targeting mechanotransduction) in the relevant sections. (Lines 359-366) (Reference 76) (4) Unfortunately, we did not find sufficient evidence linking RUNX family proteins to RPE-EMT in AMD pathogenesis; thus, this aspect was not included. We sincerely thank the reviewer for their constructive feedback, which has strengthened our manuscript.

Comments 4: Comprehensive pathway cartoons/diagrams/schematics could help summarize the well-characterized models in RPE EMT.

Response 4: We sincerely appreciate the editor's valuable suggestion regarding comprehensive pathway visualization. In response to this comment, we have taken multiple steps to enhance our manuscript's clarity: (1) We have improved Figure 2 to better summarize the key RPE-EMT signaling pathways and their crosstalk; (2) We have added a dedicated "Study Model" column in Table 1 to systematically document the experimental models used across cited references. These modifications collectively provide readers with both a clearer mechanistic understanding and improved methodological context of RPE-EMT regulation.

Comments 5: Additional discussions about emerging therapies to prevent/slow RPE EMT, such as small molecule inhibitors, gene therapy approaches, and epigenetic modulation by HDAC and BET inhibitors, would be helpful. 

Response 5: We are grateful for the editor's thoughtful suggestion to explore emerging therapeutic avenues for RPE-EMT intervention. In response, we have extensively updated our discussion on novel therapeutic strategies in Section 5. Specifically: (1) We incorporated the latest advances in small-molecule inhibitors and epigenetic modulators, organizing them by mechanistic class; and (2) To provide a more comprehensive perspective, we added three new subsections: Cell Replacement Strategies to Target RPE-EMT in AMD, Epigenetic Regulation of RPE-EMT, and Omics Landscapes of RPE-EMT. These additions highlight cutting-edge approaches while maintaining focus on translational relevance. We believe these revisions significantly strengthen the therapeutic outlook of our review.

Comments 6: Adding published transcriptomic/proteomic findings during EMT, primarily using iPSC-derived RPE disease models, would be helpful to readers interested in RPE EMT.

Response 6: We are grateful for the editor’s suggestion to strengthen the discussion of multi-omics findings in RPE-EMT. To address this suggestion, we have added a dedicated subsection titled "Omics Landscapes of RPE-EMT" in Section 5, which consolidates published transcriptomic and proteomic findings. (2) expanded "5.5. Cell Replacement Strategies to Target RPE-EMT in AMD" to link with emerging regenerative therapies. These additions create a cohesive narrative across Section 5, while addressing the editor’s emphasis on iPSC-based disease modeling.

Reviewer 2 Report

Comments and Suggestions for Authors

The manuscript's review contains important aspects. Basically, this work is well written. However, several parts should be well-revised to make the manuscript more get attentions.

1) Figure 1's late AMD (GA) and late AMD (neovascular) need to be revised. Photoreceptor cells may not die like that depicted in the image. Furthermore, CNV should be well divided with several types in AMD cases. 

2) How about HIF1a and HIF2a roles in CNV and subretinal fibrosis in AMD cases? The current research shortly discussed HIF1a only in AMD cases. HIFs should be discussed together.

3) Vitamins, minerals, or amino acids are related to RPE function. Those aspects are not discussed in this manuscript. It is important to add those aspects.

4) This work do not divide the work from animals and humans (or from mice to other species). It is important to report them in each outcome in the review article not to be confused.

5) Cell therapy parts should be well mentioned (especially, RPE cells) in the review article.

Author Response

Dear editors and reviewers:

Thank you for your letter and for the reviewers’ comments concerning our manuscript entitled “Epithelial-Mesenchymal Transition as a Key Player in AMD: Molecular Pathways and Emerging Therapies” ((Manuscript ID: biomolecules-3572242)). Those comments are all valuable and very helpful for revising and improving our paper, as well as the important guiding significance to our researches. We have studied comments carefully and have made correction which we hope meet with approval. Revised portion are marked in red in the paper. The main corrections in the paper and the responds to the reviewer’s comments are as flowing:

Responds to the reviewer’s comments:

Reviewer #2:

Comments 1: Figure 1's late AMD (GA) and late AMD (neovascular) need to be revised. Photoreceptor cells may not die like that depicted in the image. Furthermore, CNV should be well divided with several types in AMD cases. 

Response 1: We sincerely thank the editor for their critical feedback on Figure 1. In response, we have revised the depiction of late AMD subtypes to better align with clinical-pathological evidence: (1) For geographic atrophy (GA), photoreceptor degeneration is now illustrated as a progressive atrophy rather than abrupt cell death, reflecting the gradual loss observed in disease progression; and (2) In neovascular AMD (nAMD), we specifically illustrated the three types of choroidal neovascularization (CNV), each annotated with histopathological hallmarks. These revisions enhance the accuracy and educational utility of the schematic for both basic and clinical audiences.

Comments 2: How about HIF1a and HIF2a roles in CNV and subretinal fibrosis in AMD cases? The current research shortly discussed HIF1a only in AMD cases. HIFs should be discussed together.

Response 2: We sincerely appreciate the editor's insightful suggestion regarding the differential roles of HIF isoforms in AMD pathogenesis. In response, we have expanded our discussion of HIF-1α and HIF-2α in Section 3 (Lines 199-206) with the following key findings: (1) While both isoforms promote retinal neovascularization in ischemic conditions, emerging evidence (Shoda et al.) specifically implicates HIF-1α, but not HIF-2α, as a critical mediator of subretinal fibrosis; (2) Interestingly, HIF-2α appears uniquely associated with RPE metabolic dysfunction, highlighting the distinct pathological contributions of each isoform. These additions provide a more comprehensive perspective on HIF biology in AMD progression.

Comments 3: Vitamins, minerals, or amino acids are related to RPE function. Those aspects are not discussed in this manuscript. It is important to add those aspects.

Response 3: We sincerely appreciate the editor's valuable suggestion regarding the importance of nutritional factors in RPE function. In response, we have incorporated a new discussion on this critical aspect , focusing on recent breakthroughs in vitamin-mediated RPE regulation. Specifically, we highlight the work of Shen et al. demonstrating that vitamin C and valproic acid co-treatment activates mesenchymal-to-epithelial transition (MET) in fetal RPE stem-like cells through the SOX2/TGF-β1/SNAIL1 and SOX2/SNAIL1/E-cadherin pathways. This finding not only reveals novel molecular mechanisms of nutrient-mediated RPE regulation but also suggests promising applications in cell replacement therapy optimization. We have further discussed how these insights may inform future therapeutic strategies targeting RPE dysfunction in retinal diseases. In addition, after a systematic literature review, no direct evidence was found regarding mineral regulation of RPE-EMT, and thus this aspect was not included in our discussion.

Comments 4: This work do not divide the work from animals and humans (or from mice to other species). It is important to report them in each outcome in the review article not to be confused. 

Response 4: We appreciate the editor's valuable suggestion regarding the need to clarify experimental models in our review. In response, we have added a dedicated column titled "Study Model" in Table 1 to systematically categorize the experimental systems used across all cited studies, including. This modification allows readers to clearly distinguish between findings from different experimental systems while maintaining the flow of our narrative discussion. We believe this addition significantly enhances the transparency and utility of our review for researchers working across various model systems.

Comments 5: Cell therapy parts should be well mentioned (especially, RPE cells) in the review article.

Response 5: We sincerely appreciate the editor's valuable suggestion regarding the importance of cell therapy in AMD treatment. In response to this constructive comment, we have added a dedicated subsection "5.5. Cell replacement strategies to target RPE-EMT in AMD" that systematically summarizes current advances in RPE cell therapy. This new section comprehensively discusses: (1) different cell sources for RPE replacement (including iPSC-derived RPE cells), (2) key transplantation techniques, and (3) therapeutic strategies targeting RPE-EMT in cell-based therapies. We believe this addition significantly strengthens the clinical relevance of our review by highlighting cutting-edge translational approaches for AMD treatment.

Thank you for your acknowledgement of our article. We have made the corresponding changes in the article according to your suggestions, and we continue our efforts in this area.